# Smart Scalable ML-Blockchain Framework for Large-Scale Clinical Information Sharing

Anand Singh Rajawat [1], S. B. Goyal [2], Pradeep Bedi [3], Simeon Simoff [4], Tony Jan [5] and Mukesh Prasad [6,*]

1   School of Computer Sciences & Engineering, Sandip University, Nashik 422213, India
2   Faculty of Information Technology, City University, Petaling Jaya 46100, Malaysia
3   School of Computing Science and Engineering, Galgotias University, Greater Noida 203201, India
4   School of Computer, Data and Mathematical Sciences, Western Sydney University, Sydney 2751, Australia
5   Centre for Artificial Intelligence Research and Optimization, Design and Creative Technology Vertical, Torrens University, Sydney 2007, Australia
6   School of Computer Science, Faculty of Engineering and IT, University of Technology Sydney, Sydney 2007, Australia
*   Correspondence: mukesh.prasad@uts.edu.au

**Abstract:** Large-scale clinical information sharing (CIS) provides significant advantages for medical treatments, including enhanced service standards and accelerated scheduling of health services. The current CIS suffers many challenges such as data privacy, data integrity, and data availability across multiple healthcare institutions. This study introduces an innovative blockchain-based electronic healthcare system that incorporates synchronous data backup and a highly encrypted data-sharing mechanism. Blockchain technology, which eliminates centralized organizations and reduces the number of fragmented patient files, could make it easier to use machine learning (ML) models for predictive diagnosis and analysis. In turn, it might lead to better medical care. The proposed model achieved an improved patient-centered CIS by personalizing the separation of information with an intelligent "allowed list" for clinician data access. This work introduces a hybrid ML-blockchain solution that combines traditional data storage and blockchain-based access. The experimental analysis evaluated the proposed model against the competing models in comparative and quantitative studies in large-scale CIS examples in terms of model viability, stability, protection, and robustness, with improved results.

**Keywords:** clinical information sharing; blockchain; healthcare; IoT devices; consensus model; machine learning



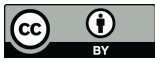

## 1. Introduction

Clinical information sharing (CIS) among medical service providers is critical for the timely treatment of patients in transit and transfer. In general, patients should be able to manage access control to their health-related data. In current practice, patients sign a paper-based or digital agreement document detailing the health-data-sharing management plans. Electronic health records (EHR) require confidentiality, integrity, and availability (CIA), and blockchain technology can be useful to protect sensitive EHR. Blockchain technology secures data using a sequence of encryption and hashing, and blockchain is considered immutable by current technology standards. Blockchain technology is a decentralized security model that can tolerate partial system failures—a useful feat for EHR. The main benefit of blockchain technology is that every transaction is dynamic and does not depend on the amount of data. Figure 1 presents a flow diagram of blockchain technology usage in a large-scale CIS.

CIS allows patients to share their information (including their current and past conditions). Both existing health records and images (which are often not encrypted) must be shared for medical examination. Figure 2 displays the schematics of the relevant blockchain model in CIS. This work proposes an innovative and improved model for sharing and

protecting health-related data using blockchain technology. The proposed model uses proxy re-encryption technologies to enhance secure data sharing among physicians in separate hospitals while patient data are protected in the blockchain.

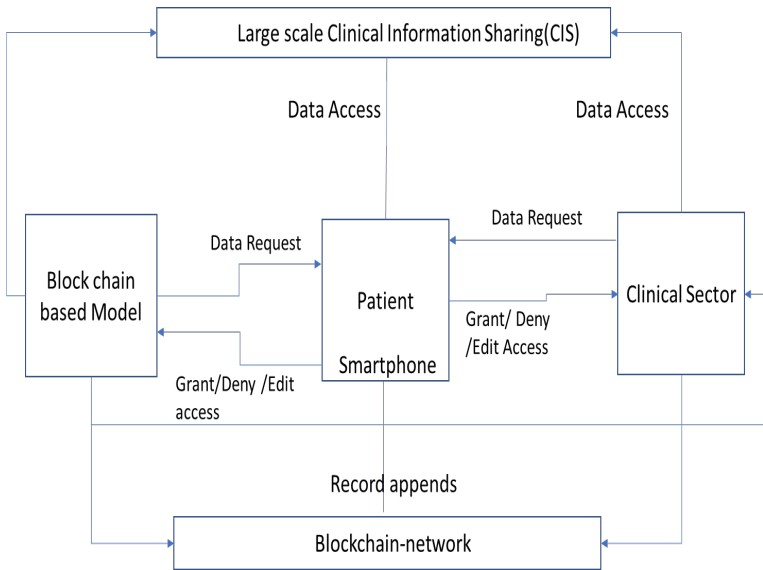

**Figure 1.** Flow diagram of blockchain modules in the healthcare industry.

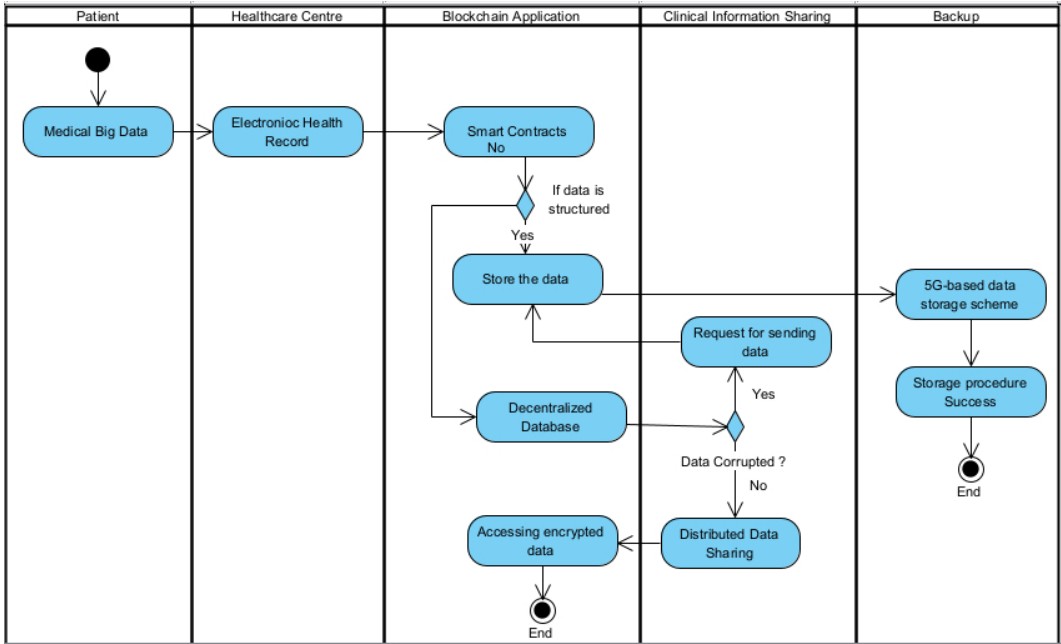

**Figure 2.** Data-flow diagram between patient, healthcare center, and distributed backup channels.

This study proposes an improved consensus mechanism that can alleviate some of the well-known concerns in medical data-sharing systems. This paper introduces a useful model that supports blockchain in medical data sharing through mutual session authentication formed between the relevant participants, which is more compact and convenient compared with the existing blockchain-enabled models. In this paper, we show how blockchain technology can be used to make it easier to share medical records. The proposed method uses decentralized networks instead of a major third party in achieving secure access control.

## 2. Related Work

The current limitation in blockchain-enabled medical data-sharing systems is that each health center must set up at least one blockchain node and complete the process of updating node(s) [1], which can hinder scalability in medical data sharing [2]. Table 1 summarizes multiple blockchain techniques used in healthcare applications.

**Table 1.** Recent blockchain approaches in the healthcare industry.

| Study | Year | Methods/Model/Algorithm | Advantage | Disadvantage | Research Gaps | Application of Blockchain in the Healthcare Informatics |
|---|---|---|---|---|---|---|
| Pustokhin, D.A. et al. [3] | 2021 | Blockchain technologies | Protect the healthcare sector | Absence of enough prototype implementation | Need to be applied to real environment | Healthcare data management |
| Sharma, L. et al. [4] | 2021 | Blockchain | Patient-centered care delivery | Provide only guidance | Need to be applied to real environment | A significant transformation in the healthcare ecosystem |
| Banotra, A. et al. [5] | 2021 | Blockchain, IoT | IoT devises measures | Data stored would be secured | Need to include machine learning | Addressing security challenges in the healthcare industry |
| Farouk, A. et al. [6] | 2020 | Blockchain, IoT | Medical data security and confidentiality violations | Secured against privacy risks | Need to apply machine learning | Different perspectives of blockchain technology in the healthcare industry |
| Musamih, A. et al. [7] | 2021 | Ethereum blockchain | Eliminating counterfeits | Limited data provenance | Need to apply machine learning | Use of blockchain in drug traceability |

The current blockchain technology can handle approximately 13–15 transactions per second. Each transaction includes authorization, touchpoint selection, and decryption key insertion/retrieval. For example, if five transactions occur per second [1], 300 CIS can be handled per minute. The blockchain processing strategy ensures that queries are delivered successfully within the constraints of scalability in the current blockchain protocol [8]. The authors [8] emphasized more on the latest mobile IoT data flow and its security (privacy in particular) requirements. In addition, it addressed in depth the latest approaches to security and privacy challenges, emerging hurdles, and future job problems. The study in [9] discussed a system for stable, accurate tracking, dissemination, and collection of EHR data. Their system preserved patient safety and maintained security in compliance with health data privacy practices, including patient access management policies [10]. The storage system features a training context synthesized by the similarities that are apparent from a few experts. The test results suggested the practicability of this approach in machine learning. The authors' [11] proposed scheme can be assured as the requested information is transmitted in a ciphertext. Blockchain and medical treatment research [12] showed that blockchain would improve the health service quality and security for the patients through its transparent principles, thereby overturning the healthcare process and establishing a modern paradigm through which patients have some input to their own treatments.

### 2.1. Healthcare Challenges during COVID-19 for Clinical Information Sharing (CIS)

Before addressing professional information exchange [13], it is essential to recognize the issues in modern healthcare systems [14]. The main objectives of healthcare service providers are to preserve lives, cure patients, and avoid delays in medical services (i.e., delays due to the repair of their machines or systems). Figure 3 shows the process-based interactions between patients, doctors/staff, and decentralized databases.

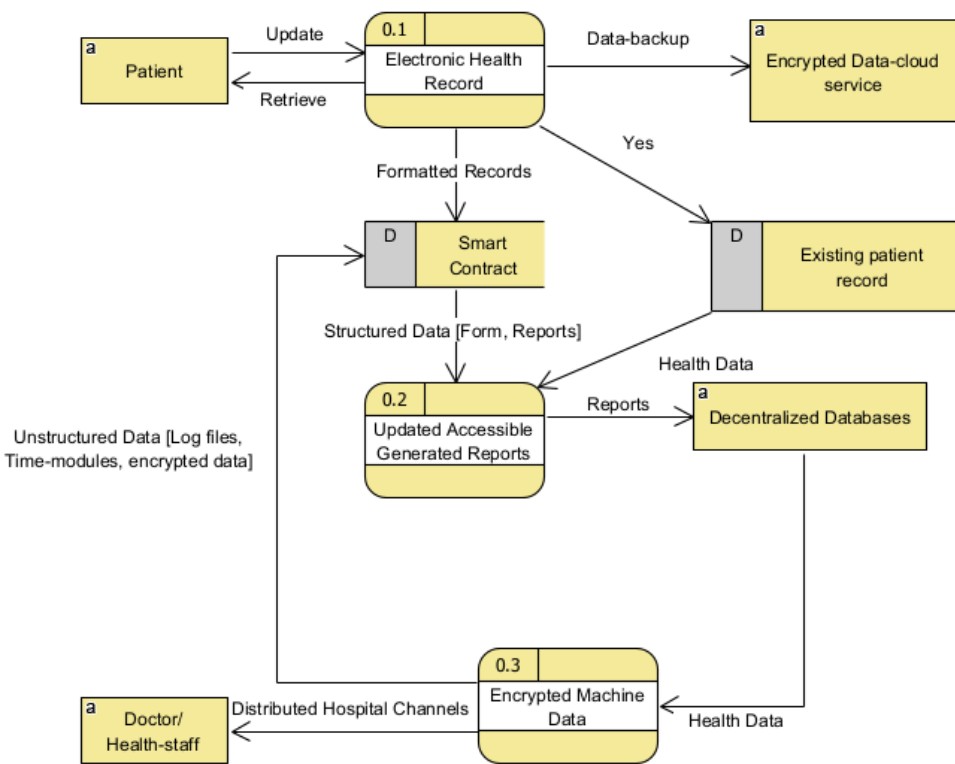

**Figure 3.** The process-based diagram between patients, doctors/staff (denoted as a), and decentralized databases (denoted as D).

During an emergency, medical practitioners need help from both Internet of Things (IoT) and traditional medical devices. IoT devices can support monitoring of patients and sharing of essential information with reduced cost overheads. However, IoT devices suffer security concerns. Linked medical equipment (from Wi-Fi to smart MRIs) increase attack surfaces for malicious actors, information-communication devices, and other data-sharing channels. Cyber attacks on IoT devices can result in security breaches and potential infringements of privacy laws for a large-scale CIS. Many legacy healthcare [12] systems are unwilling to share information without cyber protection measures.

Most medical centers do not have the IoT network compartmentalized or segmented from the other CIS appliances. As a result, any IoT device system that may be associated can have a global influence, mainly because medical and sensitive information flows laterally between devices and departments. The additional security problem is the common CIS procedures for purchasing medical devices. Security is not often stated during device acquisition or implementation, and security is often an add-on feature. The lack of built-in security capability increases cybersecurity concerns and risks to human patients. System cybersecurity concerns can include insufficient system configurations, lack of audit records, unauthorized access, and lack of device-level procedures. The following subsections examine the latest security attacks on CIS.

Table 2 displays the health data communication operation in blockchain. Table 3 depicts the landscape of potential applications in healthcare using blockchain. The usage of healthcare mobile applications during 2020 and 2021 are also shared in Table 4.

**Table 2.** Data communication operation using blockchain [15].

| Health Data | Operator | Operation |
|---|---|---|
| Health Records | Customer details, doctor records, issuance company records | Add and Update records; Query for retrieving the records |
| Medical and Health History | Medical report, check-up records, health diseases records | Add and Update records; Query for retrieving the records |
| Information of insurance | Issuance company records | Add and Update records; Query for retrieving the records |

**Table 3.** Blockchain in healthcare applications.

| Study | Year | Application of Healthcare Industry | A Key Characteristic of Blockchain |
|---|---|---|---|
| Rahmadika, S. et al. [16] | 2021 | Solitary and extensive patient records | Shared Ledger |
| Tyan, I. et al. [17] | 2021 | Principal patient files | Distributed Network |
| Kim S. [18] | 2021 | Claims adjudication | Decentralization |
| Balzarova, M.A. et al. [13] | 2020 | Supply chain management | Decentralization |
| Pan X. et al. [14] | 2020 | Interoperability | Shared Ledger |

**Table 4.** Frequently used mobile healthcare apps in 2020–2021.

| Name of the App | Description |
|---|---|
| ACT.md | Frequently used as personalized healthcare store for medical files; secure message |
| AirStrip | Delivers precise clinical information |
| ClotMD | User can manage diet library, real-time alerts |
| Doximity | Manage healthcare documents, news feed for medicine |
| Medici | Manage lab results and medical prescriptions, group chat option |

*2.2. Hidden Https Tunnels*

Attackers can hide their control connections to the medical networks in large-scale HTTPS tunnel attacks. HTTPS tunnel attacks represent remote communications (data traffic) across various sessions that appear to be normal over extended periods. If an attacker masks their command-and-control messages in HTTPS tunnels, the data traffic can masquerade as traffic from a secured service provider.

*2.3. Hidden DNS Tunnels*

The attackers using secret DNS tunnels can shield CIS exfiltration operations in healthcare networks. In a hospital, it is quite normal to move patient data. This initiates sharing of patient records between medical professionals and/or management. This may be seen as outgoing network traffic across various directions. Secret DNS tunnels are often linked to DNS touch IT and security devices. Break and catch activities are the routine operations of an IoT system. It is easier for criminals to hack and monitor the device participating in the data transfer if patient information has been identified during the transit owing to a design flaw. Additionally, the major issues related to DNS are the single point of failure risk, lack of transparency, and lack of proper DNS root-zone management.

*2.4. Ransomware and Botnet*

Although several healthcare institutions have suffered ransomware attacks in recent years, this study shows that ransomware risks were not as widespread in the second half of 2020. In addition, it is also necessary to identify and intercept ransomware attacks early before the data are maliciously hacked/encrypted and clinical procedures are interrupted.

Security and privacy protection: No one can use medical records secretly. The device should be capable of resisting disruptive threats and tracing criminal activities.

Data access: Patients will have access to some of their medical records following the authorization. Doctors may have access to previous medical information, given patient authorization.

Patient control: Patients must retain his/her personal history records—that is, without the patient's permission, no historical information can be collected.

## 2.5. Unified Standard

The unified data specification and management structure can be used by all participants in the program to help facilitate the sharing of data and encourage device stability. This research work proposes an innovative blockchain infrastructure based on the machine learning framework of [19]. Our proposed model could be one of the most economical models for healthcare systems. Every health center, clinic, or hospital connected to a decentralized system for health data sharing is considered as a separate node. First, each node in the network is discovered as a public or private address, and verification of the address of the other node is required. If a node's address is present in the approved list, the node sends a challenge message to another node. On the other hand, it provides hash key values and encrypted data linked to public addresses. If the sender node is not satisfied, it rejects the transmission and waits for the other transmissions.

## 3. Proposed Methodology

Our proposed framework based on blockchain preserves patient records and checks those who have access to data. All medical records can verify the sources of the CIS information and maintain the source of the data. Clinicians can only monitor data entry in a credible health system. Unit managers can see how many healthcare providers have achieved their results. Intelligent agreements also maintain log data and improve health information tracking. Users can only call intelligent contract functions to scan through different log file parts, depending on their locations. The primary functions of the proposed framework are as follows:

- Implement a blockchain adapter to connect to the blockchain system [20], communicate with the different nodes for sharing health records, and specify a secure interface for transmitting the information.
- Provide two levels of security to guarantee that only authorized participants are allowed to communicate smart ways to share the contract to prevent information breaches—a system for hacking to ensure data reliability.
- Ensure patient data personalization (if needed, the patient's history) only accesses the required information; for example, one can access the health information associated with a particular/specific visit to a doctor without being able to browse the complete reports.
- Perform experiments with different nodes to assess the efficacy, durability, and robustness of the proposed blockchain framework of the CIS program. It should be noted that blockchain technology is not the only approach in CIS. Investigate the applications of blockchain variants to CIS and establish guidelines and policies [21].

In the proposed blockchain, technology-driven clinical information sharing (CIS) provides a decentralized system. The proposed system provides secure CIS sharing between patients and healthcare providers including hospitals and pharmacists. In the proposed model, a patient can register and feed their details about health, which will then be converted into a hash value using the SHA 256 algorithm, and then embedded in a QR code. Using this hash value, doctors and hospitals can view the details permitted by the patients. Doctors can now provide medications by viewing the patient's records, which will be converted into a block. A pharmacist can consider this block and invoice it automatically.

### 3.1. Smart Scalable Blockchains Framework (SSBF)

SSBF uses blockchain to protect privacy. As shown in Figure 4, any transaction requires a hash with an error model. We used transaction metadata to disseminate component models and metadata (e.g., model flag, hash, and error) to merge AI to preserve privacy through a private blockchain network. We applied five flags (PREPARE (), Alteration (), Authenticate (), Interchange (), Confidentiality ()). We applied SHA 256 algorithms to

preserve stored space. All transaction operations, including SHA 256, reduce the blockchain size. First, the operation is initialized and the processing fees are set to zero to perform the transaction. At the initialization point (L = 0), each site uses available patient data to train its models and selects the model with the lowest error (User 1 with Q01 = 0.2 in our example). The justification for selecting a suitable model is to avoid errors from spreading. In this concept, we consider that the CP01 itself is "transferred" from User 1 to User 1. The chosen model (CP 01) is then presented to Users 2, 3, and 4. Afterward (L = 1), each site tests the CP 11 model using its local details. Suppose that User 2 has the greatest error (Q12 = 0.7). Although user 2 for Model CP 11 is the most unreliable, it is an example of the facts algorithm. Mts = t-time model at site s, Ets = t-time fault at site s, Mts = model at site time t, and Ets = site s time t defect. The green-subject model/error was selected at the time stamps. The exact mechanism was repeated until a consensus model (CM) [22] was identified. Note that if the E51 error is less than E54, the new data do not provide ample information for the CP54 model; so, no update or conversion is required. The same method can be applied to other locations. An additional scenario occurs when a platform leaves the private blockchain network. We do not need to deal with a situation that exists in the network based on the blockchain mechanism. If we quit the site and do not alter the model, we can forget its start. However, we can also neglect when a site exits while changing the model.

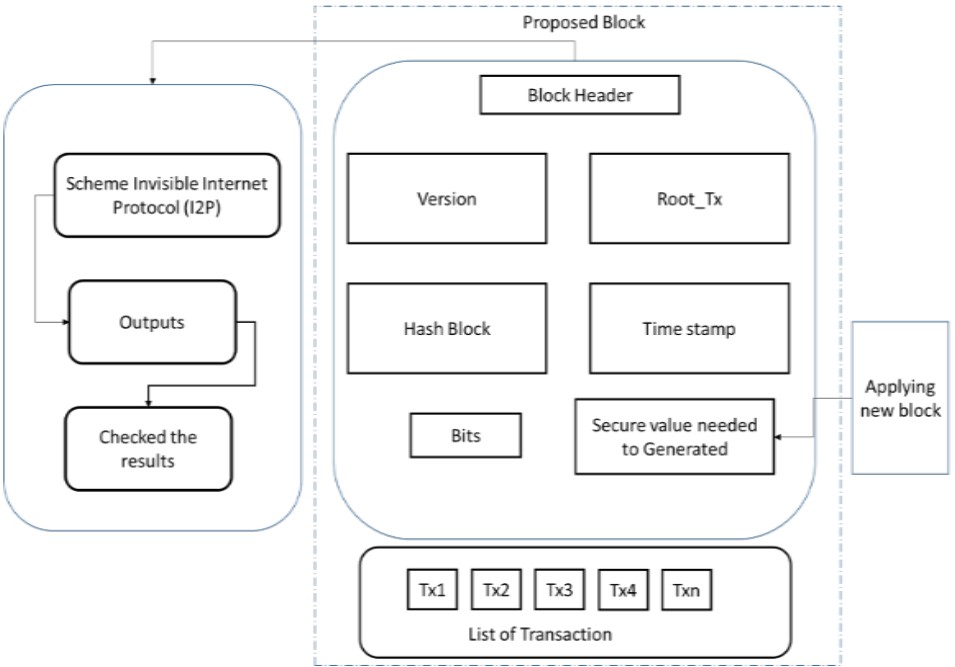

**Figure 4.** Block diagram of the blockchain-based hash function.

### 3.2. Proposed Algorithm for Large-Scale Clinical Information Sharing (LSCIS)

The user provides the input as a waiting period and canvas period, and the in-built server checks for the entire system and updates the Consensus Model (CM), which will be accessed by the user C node. The error is computed and the blockchain is checked using the CM. Additionally, the evaluation of the Consensus Model generated by the user C node and the computation [23] of the error takes place. The transaction operation is applied from the CIS, the hash function is applied with the updated signature, and the error rate ER is calculated. Upon checking the threshold value of the error rate, the transaction performed and the distributed structured [24] data are shared among different hospitals and healthcare centers.

Proposed algorithm steps:

Input: User (application used to share clinical information), canvas time period (CTP), and waiting period (WP)

Output: Advanced Al model (AIM)

- Step 1: Checking blockchain canvas time period (CTP)
- Step 2: If (applying the updated consensus model (CM))
- Step 3: Access the updated modern consensus model (CM) generated by the user C node, compute the error, and check the blockchain with the CM.
- Step 4: Evaluate the performance of the CM on Z, and calculate the error ER.
- Step 5: Perform the transaction operation CIS, apply hash function hash = HASH (CIS), and then compute the error (ER).
- Step 6: If (information is the transfer between one node and another)
- Step 7: Perform the update operation node with CIS information to produce the novel CM model to compute the new error (ER).
- Step 8: Apply and update New CM = Old CM
- Step 9: Perform the transaction operation CIS to CIS and apply hash function hash = HASH (CIS); then, compute error (ER).
- Step 10: Canvas time period (CTP), perform waiting and accumulate error rate (from other nodes).
- Step 11: Check that the ER is not higher than the error
- Step 12: Determine the node with the highest error rate.
- Step 13: Perform the transaction operation CIS to CIS, apply hash function hash = HASH (CIS), and compute the error (ER).

If any page includes new information, according to [25], it is not necessary to retrain until the consensus model is identified; if the error is less than, the new data do not provide ample information for the model; so, no update or conversion is needed. The same method can also be applied to other methods. Another scenario occurs when a platform leaves a private blockchain [26,27]. However, we can also neglect when the site exits while changing the model. This is because certain code transitions are just conceptual; the database architecture does not change until the node finishes the software update. Improvements in the model can be continued. The core algorithm for determining the learning order replaces the learning process before the consensus model is established. In Algorithm one, the CM model is applied and the second phase initializes the correct information. Select the all-new user for operation (create the new blockchain network) while the consensus learning method in the algorithm occasionally pauses before the consensus model is checked, continues, and never ends. Smart assets and contracts can be deployed automatically on every private network website. The entire transaction-based process occurs when a patient is admitted or visits a hospital, and the data are generated in diagnostic reports known as electronic health records (EHR). Blockchain technology is implemented and a smart contract is placed based on data availability from an authorized healthcare center; this also applies if the data are structured and stored through a 5G network-based [28] data storage scheme. Furthermore, in Figure 5, the data are transferred to a highly secure decentralized database, which converts the data into encrypted data that can be shared among different healthcare centers.

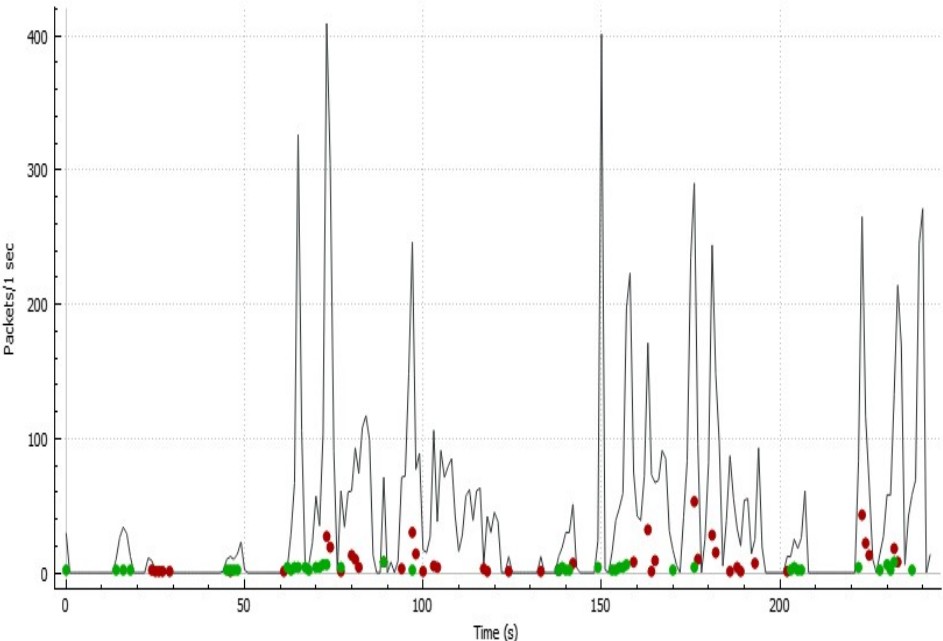

**Figure 5.** Time-series analysis of the number of packets transported per second during generation.

### 3.3. Implementation of Proposed System Registration Scheme

The registration module helps doctors and patients register their details. This module collects user details. Doctors must provide their name, date of birth, address, mail ID, phone number, and registration number.

Each doctor in the system belongs to a healthcare provider. Patients must submit their name, date of birth, address, mobile number, and details regarding their health insurance. In addition, both doctors and patients must provide valid passwords. The patients also note the details of the disease from which the patient suffers. All these details are converted into a block, and a hash value is generated. The SHA 256 algorithm is used to encrypt the values. However, no one can view these details. Even doctors cannot view patients' details until permitted by authority. The security necessities for large-scale CIS are summarized in Table 5.

**Table 5.** Explanations of security necessities for large-scale clinical information sharing (CIS).

| Necessities | Explanation |
| --- | --- |
| Freshness | Nodes cannot access outdated data |
| Non-repudiation | The sender only transmitted data for the repudiation node |
| Authentication | The sender of a received message is verified |
| Confidentiality | Data are protected against unauthorized disclosure |
| Availability | Broker node performing communication with the sensor node |
| Access control | The broker can be commutated with if the sensor node is authorized |
| Integrity | Unauthorized alteration of data can be detected |

### 3.4. Requirements for Implementing Registration Scheme

#### 3.4.1. Smart Contract Creation

Users can interact with a smart contract through an application binary interface. The user can communicate with the contractor and the doctor can send the prescribed medicines to the patients. Doctors can also conduct one-by-one sessions of reporting. The same report will be reflected on the patients, making their communication more accessible.

#### 3.4.2. Hash Value Creation

Hash Value Creation: A cryptographic hash is defined as an algorithm that takes an input and converts it into an output of a fixed value. The results are a mix of letters and

numbers. There are different types of cryptographic hashes. One such example is Bitcoin, which uses the SHA 256 algorithm. The hashing algorithm is a computational function that converts the input into a fixed-value output, called the hash value. Hash values are used to compare, identify, and run calculations against files containing data.

### 3.4.3. Block-Creation

Patients receive an EHR notification that must be approved. This method of approval avoids double-spending attacks. After verification of the CIS, the block generator module generates a block that contains CIS details. Every block in the blockchain has a separate hash of the block data.

### 3.4.4. Privacy Preservation

Blockchain technology provides a complete solution for data privacy and security. The block was connected in the form of a chain. Every block in the blockchain has a specified block header, transaction counter, and transaction. Genesis is the first block in the chain.

## 4. Results Discussion and Analysis

Data allocation time, waiting time, and total error rate are important factors in synchronous data storage and encrypted file-sharing systems in various healthcare centers. Figure 5 shows the time-series analysis of the number of packets sent over 5G-based encrypted data. The y -axis shows the number of packets transported over the 5G-based storage scheme, and the x-axis represents the time period for the entire data transfer process. In Figure 5, the green and red colored data points show the cluster points for simultaneous data back-up and encrypted data transfer for decentralized databases in clinical information sharing (CIS), respectively.

The specific field in which the current method was implemented involved the following: We used the system configuration for the simulation of 16 GB RAM with an i5 core processor @2.30GHz and Ubuntu 20.04 operating system. Python was used for cryptographic technique implementation, and simulations were performed using the PyNaCl, cryptography, and pyOpenSSL libraries. We applied 30 blockchain nodes to perform the communications. Each node was mounted on a simulation machine based on [29] and run on an independent port from the host. On an Ubuntu 20.04 computer, we developed 30 identical Ubuntu 20.04 computer labs to deploy the other 29 blockchain nodes. This computer was designed based on [30] with 16 GB RAM, an i5 core processor @2.30GHz, and Ubuntu 20.04 operating system. We designed the laboratory setup using the 16 GB RAM and Ubuntu 20.04 operating system. All systems were connected using the LAN network. The computation time for the proposed framework with dissimilar security levels are presented in Figure 6.

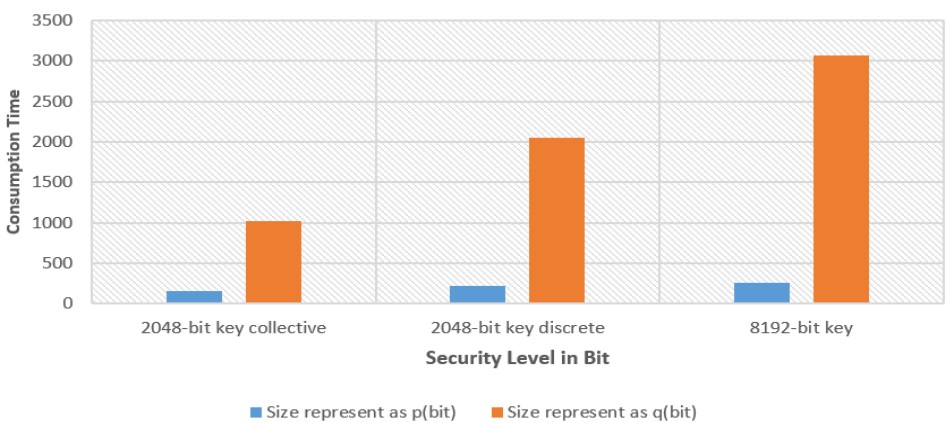

**Figure 6.** The computation time for the proposed framework with dissimilar security levels.

Public-key cryptography was used for pairing. In this study, an extra singular E(Fq) curve was used in order p above Fq, where p and q have two main numbers. Three variants (2048-bit key collective, 2048-bit key discrete, and 8192-bit key) of the advanced encryption standard (AES), RSA, and transport layer security (TLS) security key sizes were also checked. The representations of p and q in Table 6 show their respective values. Three protection parameters (k, l, l1, and 2048-bit key collective) were defined as the 2048-bit key discrete and 8192-bit key. Table 7 outlines the theoretical results of the computational cost. The results show that there were very few differences in the computational costs of the CIS services. As the security standard increases, the cost of the equipment increases gradually.

**Table 6.** Performing the simulation of different levels of security parameters p and q.

| Security Level in Bit | Size Represented as p(Bit) | Size Represented as q(Bit) |
|---|---|---|
| 2048-bit key collective | 160 | 1024 |
| 2048-bit key discrete | 224 | 2048 |
| 8192-bit key | 256 | 3072 |

**Table 7.** Simulation results at numerous security levels.

| Phase | Security Level | | |
| | 2048-Bit Key Collective | 2048-Bit Key Discrete | 8192-Bit Key |
|---|---|---|---|
| User connecting with medical facility | 3.568 | 13.334 | 26.189 |
| User connecting with medical service (connect the hospital and doctor database) | 0.004 | 0.006 | 0.006 |
| User accessing the information from the hospital database | 3.366 | 11.259 | 25.150 |
| User connecting with the insurance company database | 1.956 | 1.892 | 1.902 |
| User connecting with another hospital and doctor database | 9.449 | 48.149 | 120.742 |

Figures 7–9 represent the costs of networking for patients and clinicians—that is, data transmission, data authentication, and data search and access, which are considered valuable phases. In the proposed scheme, C is the customer Clm patient necessity sending the (C1, C2) trapdoor to the local server, where C1 and C2 are G1 components. If the doctor has to check for a record of historical use, the private key KM would be submitted to KM as an element of E−p. For Doctor D, the private key KD must be sent to the DM, obtaining the encrypted record kept in another hospital, where KD is part and the cipher text (CT) of the historical record (HD) is similar to Clm. In the blockchain (BC), where BC1, BC2, and BC5 are the elements of CP1, BC3 is the element of size, BC4 is the product of size L, the username of the pseudo-identification is the element of the general ciphertext space, and Doctor D is responsible for transmitting BClm = (BC1, BC2, BC3, BC4, BC5). This is our communications cost, where np represents the number of verifiers of the blockchains, which is private, and the time stamp is described as t. Therefore, the planned structure is stronger, and finally, to display the contact prices, 160 bits and 1024 bits are set for p and q. The components of CP1 and CP2 are 1024 bits long and are separated from 512 bits. We presume that all 32 bits, 160 bits, np = 3, and 256 bits are the identifier, label time, and the ciphertext space.

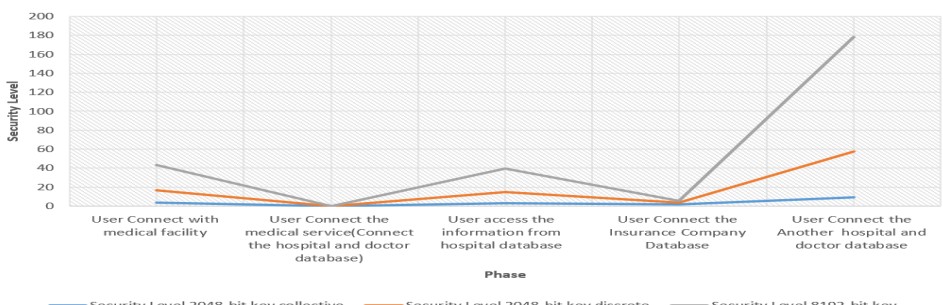

**Figure 7.** Comparison of security threshold level and different phases at the hospitals.

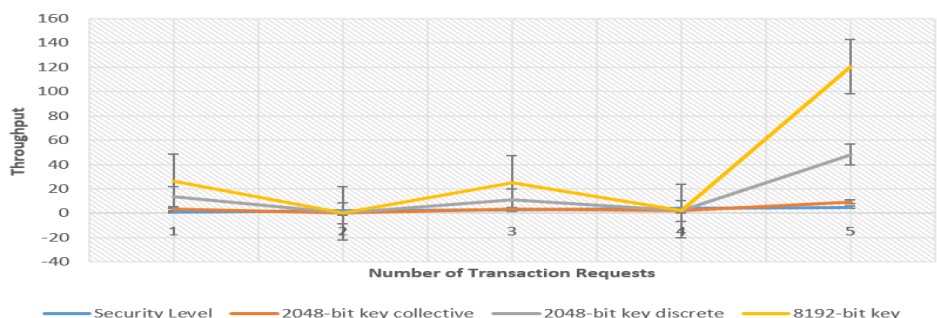

**Figure 8.** Comparison of throughput without security countermeasures.

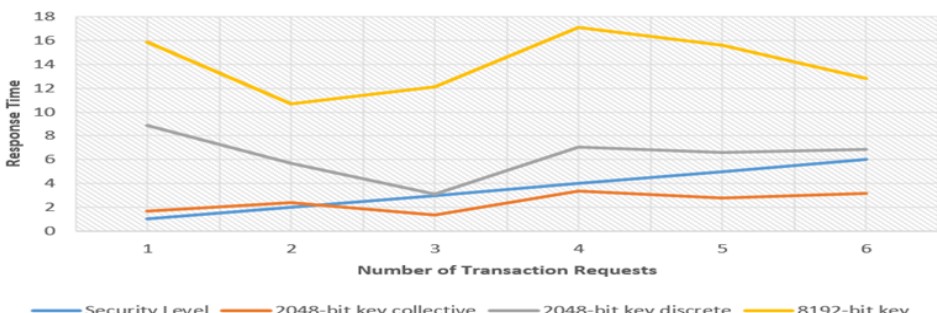

**Figure 9.** Comparison of response time without security countermeaures.

## 5. Conclusions

This study introduces critical technology components for a number of organizations to share patient data in blockchain-based CIS. The prototype demonstrated the feasibility of our approach. Each hospital had a node integrated into a blockchain network. This study adopted an intelligent method in which only metadata maintenance was stored in the chain, unlike the other models in which the CIS details are embedded and stored off-chain. The proposed system utilizes asymmetric encryption and digital signatures that rely on public key networks to ensure the availability of standard clinical information in CIS. A set of comprehensive evaluation and comparative testing analyses showed that the proposed model provides a lightweight blockchain solution that has been proven useful against the other competing models in large-scale CIS applications. Our proposed model uses the User to Connect with a medical facility (2048-bit key collective, 2048-bit key discrete, 8192-bit key) (3.568, 13.334, 26.189). The ability to store cryptographic records securely is a key feature of blockchain technology that is essential for ML. This would enable the institution/provider-based networks to grow and evolve, simultaneously allowing them to connect to similar networks. This notion of aggregation, or nested blockchains, may be an approach to extending the reach of collaborations and sharing beyond local networks.

**Author Contributions:** Conceptualization, A.S.R. and S.B.G.; methodology, A.S.R. and P.B.; software, A.S.R.; validation, S.B.G., S.S., T.J. and M.P.; formal analysis, A.S.R. and P.B.; investigation, A.S.R. and S.B.G.; resources, S.B.G., T.J. and M.P.; data curation, A.S.R. and S.B.G.; writing—original draft preparation, A.S.R. and P.B.; writing—review and editing, A.S.R., S.B.G., P.B., S.S., T.J. and M.P.; visualization, A.S.R. and P.B.; supervision, S.B.G. and M.P.; project administration, S.B.G. and P.B.; funding acquisition, S.S., T.J. and M.P. All authors have read and agreed to the published version of the manuscript.

**Funding:** This research received no external funding.

**Institutional Review Board Statement:** Not applicable.

**Informed Consent Statement:** Not applicable.

**Data Availability Statement:** Not applicable.

**Conflicts of Interest:** The authors declare no conflict of interest.

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
