# Peer review of "Smart Scalable ML-Blockchain Framework for Large-Scale Clinical Information Sharing"

_applsci, doi:10.3390/app122110795_

Round 1

Reviewer 1 Report

Dear Authors,

The attached review letter contains my comments, questions, and suggestions.

Kind regards,

Reviewer 2 Report

1- the abstract lack the motivation of the study 

2- introduction is missing the contributions of the study add them at the end of the section 

3- add numerical results to conclusion 

4- enrich the work by citing recent published work 

Israa Al_Barazanchi, Aparna Murthy, Ahmad AbdulQadir Al Rababah, Ghadeer Khader, Haider Rasheed Abdulshaheed, Hafiz Tayyab Rauf, Elika Daghighi, & Yitong Niu. (2022). Blockchain Technology - Based Solutions for IOT Security. Iraqi Journal For Computer Science and Mathematics3(1), 53–63. https://doi.org/10.52866/ijcsm.2022.01.01.006

Sabah, noor, Sagheer, A., & Dawood, O. (2021). Survey: (Blockchain-Based Solution for COVID-19 and Smart Contract Healthcare Certification). Iraqi Journal For Computer Science and Mathematics2(1), 1–8. https://doi.org/10.52866/ijcsm.2021.02.01.001

5- moreover there are many grammatical and formatting errors 

Round 2

Reviewer 1 Report

Accept in present form